# Optimization of the Green Fibre Paper Film Preparation Process Based on Box–Behnken Response Surface Methodology

Xiaoqing Cao [1], Lu Li [1,*], Fengwei Zhang [1], Linxiong Shi [1,2], Fangyuan Zhang [1], Xuefeng Song [1], Wuyun Zhao [1] and Fei Dai [1,*]

1   College of Mechanical and Electrical Engineering, Gansu Agricultural University, Lanzhou 730070, China; caoxq@st.gsau.edu.cn (X.C.); zhangfw@gsau.edu.cn (F.Z.); gsslx01@126.com (L.S.); zfangy@st.gsau.edu.cn (F.Z.); songxf@gsau.edu.cn (X.S.); zwy@gsau.edu.cn (W.Z.)
2   Gansu Agricultural Mechanization Technology Extension Station, Lanzhou 730046, China
*   Correspondence: lilu@gsau.edu.cn (L.L.); daifei@gsau.edu.cn (F.D.); Tel.: +86-0931-7631207 (L.L.); +86-0931-7677809 (F.D.)

**Abstract:** To improve the utilization rate of flax straw and the clean treatment of livestock manure, an experimental study was conducted on the process and performance of making fibre paper films by mixing cow dung and flax straw fibre. Cow dung and flax straw were used as the main raw materials, and functional additives were not added. The whole technological process of the pretreatment, the beating process, the determination of the beating degree, the basis weight of the paper, papermaking, drying, sample cutting, and the determination and analysis of the related mechanical properties of the film-making materials were studied. In this study, the Box–Behnken experimental design principle in the response surface methodology was adopted, and the effect of each factor on the tensile strength and tear strength of fibre paper film made of mixed fibres was determined using the combined experimental design comprising four factors and three levels centres. The results showed that the optimum technological parameters were as follows: the beating degree of the cow dung fibre was 37 °SR, the beating degree of the flax straw fibre was 85 °SR, the paper basis weight was 80 g/m$^2$, and the addition of flax straw fibre was 65%. At a drying temperature of 80 °C and a drying time of 8 min, under the conditions of the hybrid fibre paper film placed in the laboratory environment (humidity of 30%~40%, temperature of 18 °C) for 24 h, the measured tensile strength was about 8.26 MPa, and the tear strength was about 19.91 N/mm. This study provides a reference that can be used for the further study of fibre paper film.

**Keywords:** cow dung; fibre paper film; flax straw; response surface methodology

## 1. Introduction

Plastic film mulching is an agricultural production and cultivation technology that is widely used in arid areas and suitable for different crops [1,2]. Plastic film is widely used as a water-saving measure in agriculture worldwide [3]. In recent years, the widespread coverage and cumulative use of plastic film have caused serious environmental problems [4–7]. The residual plastic film that has accumulated in farmland soil and fields caused severe environmental problems and the destruction of soil structure [8], affecting the healthy development of the ecological environment and normal crop growth. To solve the pollution problem of plastic film, a variety of degradable mulch films has been actively studied, such as photodegradable mulch film, biodegradable mulch film, starch mulch film, liquid mulch film, and fibre paper film [3,9–13].

In recent years, plant fibres have attracted significant attention from scholars and have become a research hotspot in the field of "green materials". Plant fibre mainly comes from crop straw, forage, wood, etc. [14], and has the characteristics of having abundant resources, wide sources, environmental protection, and good degradability and being pollution-free.

Therefore, using plant fibres to develop a green fibre paper film instead of non-degradable plastic film is one way to solve environmental pollution problems.

In recent years, with the adjustment of industrial structure, the improvement of people's quality of life, and the increasing demand for animal products, the large-scale development of animal husbandry has been promoted. As the cattle industry is an important part of animal husbandry, the scale of the cattle industry has rapidly expanded, bringing economic benefits while also generating a large amount of fecal pollution. The accumulation of fecal pollution and the generation of foul gases have caused environmental pollution to rise in rural areas [15]. The rational treatment of cow dung has become an urgent problem to be solved. In this paper, cow dung and flax straw fibre were used as raw materials to prepare the degradable fibre paper film. Among them, flax, as a high-quality oil crop, has been widely planted both domestically and internationally. But, flax straw is mostly burned as fuel in rural areas; it has not been better utilized and, instead, pollutes the environment.

Related studies have shown that cow dung and flax straw have high fibre contents [16,17] and are abundant in resources, which are factors that make them suitable for preparingfibre paper films. For cow dung, Yang et al. [15] used different treatment methods to extract cellulose from cow dung to make the paper film. The results showed that the extraction rate of cellulose from cow dung treated with KOH is high (42%) and the paper performance (a burst index of 2.48 Kpa·m$^2$/g, a tear index of 4.83 mN·m$^2$/g, and a tensile index of 26.72 Nm/g) are exceptional. Compared with single-phase materials, composites show better performance, such as their supreme microstructures, morphology, and mechanical properties [18]. Chaturvedi et al. [19] prepared a novel polymer composite film composed of cow dung fibres and polyvinyl alcohol (PVA) using a hand lay-up method. Vedrtnam [20] manufactured a novel polymer composite film using cow dung fibres (soaked in 5% NaOH solution) and polyvinyl alcohol (PVA). These two results showed that the alkali-treated cow-dung-fibre-composite (CDFC) film had increased bounding and a reduced fibre pullout, resulting in superior mechanical properties. Fasake et al. [21] used a mixture of cow dung fibre and cotton pulp to prepare the paper film, and the results showed that paper with the desired level of quality could be made by removing lignin from cow dung and adding some mixed materials with higher cellulose. There are also studies showing that an increase in cow dung concentration in the cow dung cardboard causes it to exhibit antibacterial activity, which means it can be used for different packaging materials [22,23]. After waxing, this paper material can be used for wet and fatty foods [24]. For flax straw, Bodros et al. [25] showed that biodegradable L-polylactide acid (PLLA)/flax fibre mat composites exhibiting specific tensile properties equivalent to glass fibre polyester composites can be manufactured by an unoptimised film-stacking process. Avci et al. [26] used an extrusion process to produce biocomposites based on polylactic acid (PLA) and short flax fibres. The research results showed that, with the addition of flax fibres, the thermal stability of PLA was improved, and the flax fibres added to PLA, along with the coupling agent, improved the tensile strength, the bending strength, and the bending modulus. Therefore, cow dung and flax straw have certain advantages as raw materials for fibre paper films.

In this study, we proposed a renewable and directly recyclable plant fibre paper film for agricultural mulching using cow dung and flax straw as raw materials, which are green and cheap, and have abundant sources. By using the Box–Behnken experimental design principle in the response surface methodology and the analysis method of the quadratic regression model, the optimal combination of process parameters for preparing fibre paper film by mixing flax straw fibre and cow dung fibre was studied. By analyzing the effects of factors such as the cow dung fibre beating degree, the flax straw fibre beating degree, the paper basis weight, and the addition amount of flax straw fibre on the properties of fibre paper film, the aim of this study was to determine a reliable set of preparation process parameters, and provide a parameter basis for the subsequent preparation of paper films

made from cow dung, flax straw, and other straw fibres, in order to explore the feasibility of comprehensively utilizing flax straw and cow dung to produce fibre paper films.

## 2. Materials and Methods

### 2.1. Materials and Equipment

#### 2.1.1. Experimental Materials

Flax straw fibre raw material (Variety: Dingya No. 23, with Flax content of 40%) was purchased from Xizhai Oil Testing Station in Dingxi City, Gansu Province, China and extracted by mechanical rolling. Cow dung was from Huarui Agricultural Company in Minle Ecological Industrial Park, Zhangye City, Gansu Province, China.

#### 2.1.2. Test Instruments and Equipment

We used a 500A crusher (Dongguan Fangtai Electric Appliance Co., Ltd., Dongguan, China) to crush the raw materials. For the preparation of pulp, we used the TD1-15 experimental digester (Xianyang Tongda Light Industry Equipment Co., Ltd., Xianyang, China) to cook the flax straw, then used the TD6-23 Wali Pulping Machine (Xianyang Tongda Light Industry Equipment Co., Ltd., Xianyang, China) to beat the pulp, and the TD9-M beating degree tester (Xianyang Tongda Light Industry Equipment Co., Ltd., Xianyang, China) to measure the beating degree. For the preparation of paper film samples, we used the TD15-A standard defibrizer (Xianyang Tongda Light Industry Equipment Co., Ltd., Xianyang, China) for fibre unwinding, and then used the TD10-200 paper sheet former (Xianyang Tongda Light Industry Equipment Co., Ltd., Xianyang, China) to complete the paper film forming. In order to determine the mechanical strength of the fibre paper film sample, we used the BY-4012B pneumatic punching machine (Yangzhou Boyu Testing Machinery Factory, Yangzhou, China) to cut the sample, and then used the CMT8102 microcomputer controlled electronic universal testing machine (accuracy level: 0.5; Shenzhen Xinsansi Material Testing Co., Ltd., Shenzhen, China) to determine the mechanical strength of the fibre paper film sample. We used the S-3400N scanning electron microscope (Suzhou Sainz Instrument Co., Ltd., Suzhou, China) to observe the microstructure of the sample.

### 2.2. Experimental Procedure

The preparation flow chart is shown in Figure 1:

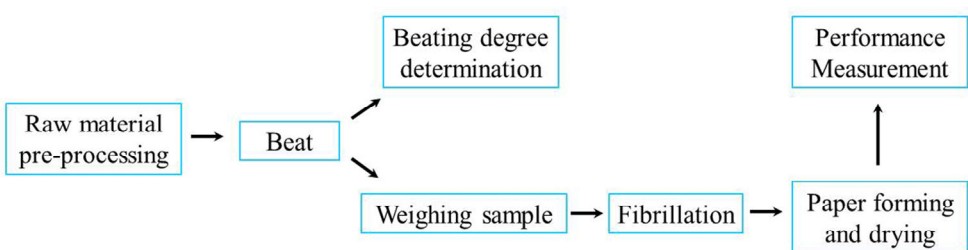

**Figure 1.** Preparation Flow Chart.

(1)    Fibre extraction

The purchased original flax straw was mechanically rolled to remove some wood components to achieve, a flax content of about 40%. We divided 1500 g of the purchased cow dung raw materials evenly into three plastic buckets (10 L), immersed the cow dung in tap water, soaked it for 2 h, and then cleaned it to remove impurities from the cow dung. Then, the dung was completely dried at 65 °C for subsequent pulping.

(2)    Cooking of flax straw

The treated flax straw fibres were weighed out to 400 g and soaked in tap water at room temperature for 8 h. The fibres were then placed in a laboratory digester (15 L), and then steamed for 30 min at 160 °C, which was designed to destroy the lignin, soften the fibres, and improve the cellulose yield.

(3)    Beating

①  Flax straw fibre

The cooked dry flax straw fibre was soaked in clear water for about 10 h before beating according to the required concentration (1.34 g/mL), and then was beaten with a TD6-23 Kawara Tsutomu beater. As flax straw fibre is easy to twine into a cluster, it was necessary to first loosen fibre without any load for about 30 min, then gradually increase the load to complete the beating process. During the beating process, we took pulp with different beating degrees for later use.

②  Cow dung fibre

As with the beating process of flax straw as mentioned above, a certain amount of dried cow dung was weighed according to a certain concentration (3.25 g/mL) before beating, and it was soaked in clear water for about 10 h. Then, it was beaten with load, and pulps with different beating degrees were taken for subsequent tests.

③  Determination of beating degree. Refer to the standard GB/T 3332-2004 [27].

(4)    Paper film forming via Rapid Kaiser Method

The used pulp was dried at 55 °C for 2 h, then weighed and mixed once absolutely dry according to different ratios and the paper basis weight. Tap water (at room temperature)was added to immerse the pulp. After the pulp was completely dissociated, the fibres were put on a paper former for paper forming and drying (drying temperature 80 °C).

(5)    The mechanical properties of the fibre paper film samples

In this experiment, the tensile and tear strength of the fibre paper film were measured. Dumbbell-shaped samples were used in the tensile test, and the samples were cut with a special cutting machine. The standard distance of the samples was 25 mm, the width was 6 mm, and the length was 115 mm. The tear test sample was a right-angle tear sample, which was also cut by a cutting machine, and the sample length was 100 mm. The samples are shown in Figure 2.

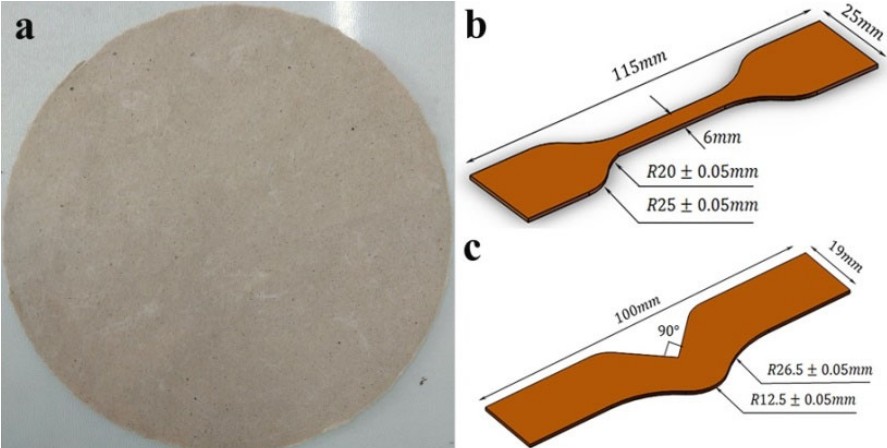

**Figure 2.** Macroscopic photos of (**a**) the fibre paper sample, (**b**) tensile sample and (**c**) tear sample. (R in the figure represents the radius at the corner of the sample, where R20, R25, R26.5 and R12.5 denote the radii of 20, 25, 26.5, and 12.5, respectively, in mm, with an error of ±0.05).

The mechanical properties of the fibre paper film samples were determined after being left for about 24 h at room temperature (18 °C) and air humidity of 30%~40%. According to GB/T 35795-2017 [28], the tensile strength and tear strength of the fibre paper film were measured. Each group of experiments was repeated 8 times, the data were optimized and analysed by using Design-Expert 8.0.6 software, and the data were statistically analysed by using Origin 9.8.0.

Tensile strength:

$$\sigma = \frac{F_1}{A} \tag{1}$$

$\sigma$—Tensile strength, MPa;
$F_1$—Tensile breaking load, N;
$A$—Section surface, mm$^2$.
Tear strength:

$$T_s = \frac{F_2}{t} \tag{2}$$

$T_s$—Tear strength, N/mm;
$F_2$—Tearing load, N;
$t$—Thickness, mm.

### 2.3. Experimental Design

2.3.1. Single-Factor Experimental Design

Through the analysis of the preparation process of paper film, it was determined that the main factors affecting the performance of the mixed film of cow dung and flax straw fibre are: the beating degree of flax fibre, the additional amount of flax straw fibre, the beating degree of cow dung fibre, and the paper basis weight. The beating degree of cow dung fibre (33, 37, 41, 45, 49 °SR), the beating degree of flax straw fibre (73, 77, 81, 85, 89 °SR), and the paper basis weight (50, 60, 70, 80, 90 g/m$^2$) were studied.

2.3.2. Design of Response the Surface

In this experiment, the Box–Behnken experimental design principle in Response Surface Methodology was adopted. The beating degree of cow dung fibre, the beating degree of flax straw fibre, the paper basis weight, and the amount of flax straw fibre added were taken as effecting factors. The tensile strength (MPa) and tear strength (N/mm) were taken as performance evaluation indexes. A combined experimental design with four factors and three levels was carried out to analyse the effect of each factor on the strength of the fibre paper film. According to the experimental design, a total of 29 groups of experiments were conducted, with 4 fibre paper films with the same factor prepared for each group. A total of 116 fibre paper film samples were prepared. In each group of experiments, 8 repeated experiments were conducted for each indicator, and the average value of the 8 repeated tests was used as the experimental result for each evaluation indicator. All evaluation indicators were based on the average of 8 replicates. All evaluation indexes were based on the average value of 8 repetitions. The tensile strength and tear strength characterize the maximum force at which the fibre paper film ruptures under longitudinal load. The encoding table for effecting factor levels is shown in Table 1.

**Table 1.** Encoding table for effecting factor levels.

| Levels | Factors | | | |
|---|---|---|---|---|
| | **Beating Degree of Cow Dung/°SR** | **Beating Degree of Flax Straw/°SR** | **Paper Basis Weight/(g/m$^2$)** | **The Addition Amount of Flax Straw Fibre/%** |
| −1 | 33 | 81 | 70 | 55 |
| 0 | 37 | 85 | 80 | 65 |
| +1 | 41 | 89 | 90 | 75 |

Note: In the table, the addition amount of flax draw fibre represents the mass proportion of flax straw fibre in the 80 g/m$^2$ fibre paper film.

## 3. Results and Analysis

### 3.1. The Effect of Various Factors on the Tensile and Tear Strength of the Fibre Paper Film

3.1.1. Effect of Beating Degree of Cow Dung on the Tensile and Tear Strength of the Fibre Paper Film

The controlled variable method was used to prepare the paper fibre film from cow dung pulps with different beating degrees, and the strength of the paper fibre film was determined. The beating degree of cow dung has a certain effect on the tensile strength and tear strength of the fibre paper film, and the impact on tear strength is more obvious. The test results are shown in Figure 3a. As shown in Figure 3a, as the beating degree of cow dung fibre increases, the tensile strength of the paper fibre film increases firstly and then decreases, and finally tends to a stable state. When the beating degree of cow dung fibre is around 37 °SR, the tensile strength of the fibre paper film is significantly higher than that of other beating degrees. The tear strength also shows the same trend: with the increase in the beating degree of cow dung fibre, the tear strength first increases and then decreases, and the changing trend is more obvious. The tear strength is highest at around 37 °SR. Before the beating degree of cow dung fibre reached 37 °SR, the tensile strength and tear strength showed obvious changes with the increase in the beating degree. From the scanning electron microscopy image of the fibre paper film surface in Figure 4a, it can be observed that at low beating degrees, the cow dung pulp was relatively coarse, and the cellulose wrapped in lignin was not exposed. Therefore, after mixing with flax straw fibre, there was no better bonding between two types of fibres, and the structure between the paper film fibres was not tight, resulting in lower strength. As the beating degree of cow dung increases, the wooden structure on the fibre surface is destroyed, and the cellulose is exposed. The bonding effect between two types of fibres improves, and the strength increases. Finally, as the beating degree further increases, the strength of the fibre paper film decreases. From Figure 4b, it can be observed that during the beating process, as the beating time increases, the physical structure of the cow dung fibre is destroyed, resulting in a decrease in strength.

3.1.2. Effect of Beating Degree of Flax Straw Fibre on the Tensile and Tear Strength of the Fibre Paper Film

The beating degree of flax straw fibre has a certain effect on the strength of the fibre paper film. In the experiment, the beating degree of cow dung fibre, the paper basis weight, and flax straw fibre addition were taken as the specified factors, and the beating degree of flax straw fibre was taken as the changing factor to analyse the effect of flax straw beating degree on the strength of the fibre paper film. The experimental results are shown in Figure 3b.

From Figure 3b, it can be seen that as the beating degree of flax straw fibre increases, the tensile strength and tear strength of the fibre paper film show a trend of first increasing and then decreasing. When the beating degree of flax straw fibre is around 81 °SR, the tensile strength and tear strength are higher than those under other beating degrees. However, with the increase in beating degree (73 °SR–89 °SR), the trend of changes in tensile strength and tear strength is not particularly significant. When the beating degree is below 81 °SR, as the beating degree increases, the degree of brooming of flax straw fibre increases, and the bonding effect among fibres is significant. The tensile strength and tear strength of the fibre paper film show an increasing trend. Moreover, at low beating degrees (around 73 °SR), the brooming degree of flax straw fibre is low, and the fibre distribution in the fibre paper film is uneven, resulting in poor overall quality of the fibre paper film. The impact of uneven fibre dispersion on the quality of the fibre paper film can be observed in the SEM image in Figure 4c, resulting in lower tensile and tear strength of the fibre paper film. When the beating degree is high (>81 °SR), the tensile strength and tear strength of the fibre paper film decrease. This is because as the beating degree increases, the structure of a single fibre is destroyed, leading to a decrease in fibre adhesion and a decrease in paper film strength.

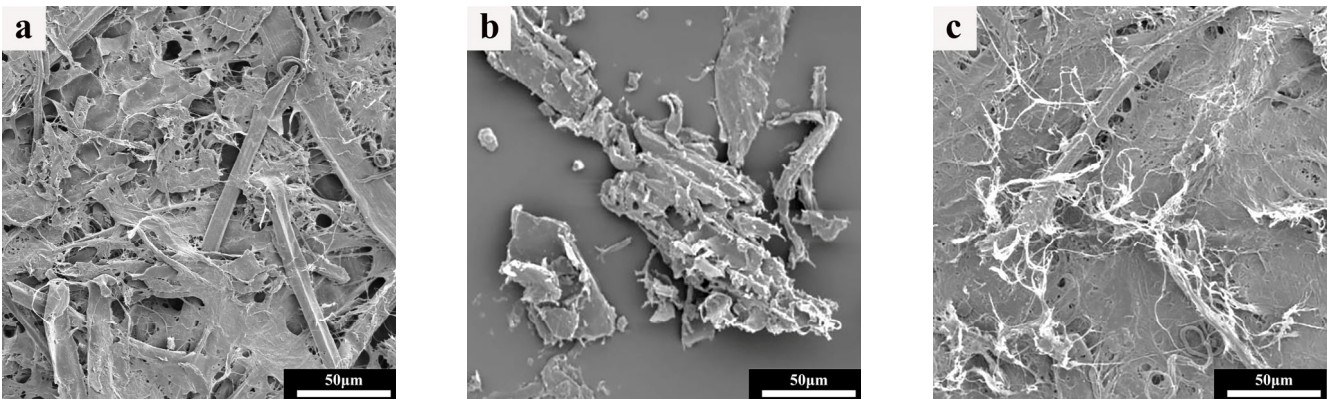

**Figure 3.** Effect of (**a**) the beating degree of cow dung fibre, (**b**) the beating degree of flax straw fibre, (**c**) the paper basis weight, and (**d**) the addition amount of flax straw fibre on the strength of the fibre paper film.

**Figure 4.** SEM images (×500 times) (**a**) low beating degree fibre paper film for cow dung, (**b**) cow dung pulp, (**c**) low beating degree fibre paper film for flax straw.

### 3.1.3. Effect of the Paper Basis Weight on the Tensile and Tear Strength of the Fibre Paper Film

The paper basis weight has a certain effect on the strength of the fibre paper film [29]. In the experiment, the beating degree of cow dung fibre, beating degree of flax straw fibre and flax straw fibre addition were taken as the specified factors, and the paper basis weight was taken as the changing factor to analyse the effect of the paper basis weight on the strength of the fibre paper film. The test results are shown in Figure 3c.

As can be seen in Figure 3c, with the increase in the paper basis weight, the tensile strength of the fibre paper film first increases gradually and then tends to be stable, and the tear strength increases with the increase of the paper basis weight. In Figure 3c, it can be seen that the change of tensile strength is not obvious, because with the increase in the paper basis weight, the thickness of the fibre paper film increases, the cross-sectional area of the tested sample increases, and the tensile strength of the fibre paper film changes slightly. However, the tear strength shows an obvious change, because with the increase in the paper basis weight, the fibre amount per unit area of the fibre paper film increases, and its tear strength becomes stronger under the comprehensive force of fibres [30].

### 3.1.4. Effect of Flax Straw Fibre Addition on the Tensile and Tear Strength of the Fibre Paper Film

The content of flax straw fibre has a significant effect on the strength of fibre paper film, because flax straw fibre has good mechanical strength and is a high-quality plant fibre [31]. During the experiment, the beating degree of cow dung fibre, beating degree of flax straw fibre, and paper basis weight were taken as the specified factors to analyse the effect of the addition of flax straw fibre on the strength of the fibre paper film. The test results are shown in Figure 3d.

It can be clearly seen in Figure 3d that with the increase in flax straw fibre content, the tensile strength and tear strength of the fibre paper film increase rapidly, and the changing trend of tear strength is more obvious. However, with the increase in flax straw fibre content, the amount of cow dung decreases, and the cost of the fibre paper film preparation increases, so the production cost should be reduced as much as possible in the actual production process.

### 3.2. The Design Scheme and Result Analyses of Response Surface Methodology

The test results are shown in Table 2:

**Table 2.** Test Plan and Results.

| Test Number | Beating Degree of Cow Dung/°SR | Beating Degree of Flax Straw/°SR | Paper Basis Weight/(g/m²) | The Addition Amount of Flax Straw Fibre/% | Tensile Strength/MPa | Tear Strength/ (N/mm) |
|---|---|---|---|---|---|---|
| 1 | −1 | −1 | 0 | 0 | 4.70 | 13.30 |
| 2 | 1 | −1 | 0 | 0 | 4.52 | 13.01 |
| 3 | −1 | 1 | 0 | 0 | 5.92 | 14.13 |
| 4 | 1 | 1 | 0 | 0 | 5.73 | 12.06 |
| 5 | 0 | 0 | −1 | −1 | 4.74 | 13.57 |
| 6 | 0 | 0 | 1 | −1 | 6.57 | 14.81 |
| 7 | 0 | 0 | −1 | 1 | 8.01 | 18.61 |
| 8 | 0 | 0 | 1 | 1 | 9.06 | 22.33 |
| 9 | −1 | 0 | 0 | −1 | 4.29 | 11.58 |
| 10 | 1 | 0 | 0 | −1 | 4.24 | 12.11 |
| 11 | −1 | 0 | 0 | 1 | 6.55 | 18.74 |
| 12 | 1 | 0 | 0 | 1 | 6.32 | 17.56 |
| 13 | 0 | −1 | −1 | 0 | 5.37 | 13.99 |
| 14 | 0 | 1 | −1 | 0 | 8.23 | 13.76 |

**Table 2.** *Cont.*

| Test Number | Beating Degree of Cow Dung/°SR | Beating Degree of Flax Straw/°SR | Paper Basis Weight/(g/m²) | The Addition Amount of Flax Straw Fibre/% | Tensile Strength/MPa | Tear Strength/ (N/mm) |
|---|---|---|---|---|---|---|
| 15 | 0 | −1 | 1 | 0 | 7.86 | 17.49 |
| 16 | 0 | 1 | 1 | 0 | 8.59 | 16.00 |
| 17 | −1 | 0 | −1 | 0 | 5.24 | 15.68 |
| 18 | 1 | 0 | −1 | 0 | 5.12 | 16.03 |
| 19 | −1 | 0 | 1 | 0 | 6.87 | 19.05 |
| 20 | 1 | 0 | 1 | 0 | 6.57 | 16.15 |
| 21 | 0 | −1 | 0 | −1 | 5.16 | 11.03 |
| 22 | 0 | 1 | 0 | −1 | 5.88 | 12.32 |
| 23 | 0 | −1 | 0 | 1 | 6.59 | 18.96 |
| 24 | 0 | 1 | 0 | 1 | 8.74 | 15.66 |
| 25 | 0 | 0 | 0 | 0 | 7.08 | 19.29 |
| 26 | 0 | 0 | 0 | 0 | 7.55 | 18.98 |
| 27 | 0 | 0 | 0 | 0 | 7.87 | 19.91 |
| 28 | 0 | 0 | 0 | 0 | 8.11 | 20.09 |
| 29 | 0 | 0 | 0 | 0 | 7.88 | 18.99 |

*3.3. Regression Model*

Design-Expert 8.0.6 was used to analyse the test results in Table 3, and the quadratic regression models of tensile strength $y_1$ (MPa) and tear strength $y_2$ (N/mm) were obtained, as shown in Formulas (3) and (4).

$$y_1 = 7.70 - 0.089x_1 + 0.74x_2 + 0.73x_3 + 1.20x_4 - 0.0025x_1x_2 - 0.045x_1x_3 - 0.045x_1x_4 - 0.53x_2x_3 \\ + 0.36x_2x_4 - 0.19x_3x_4 - 1.88x_1^2 - 0.47x_2^2 + 0.14x_3^2 - 0.62x_4^2 \tag{3}$$

$$y_2 = 19.45 - 0.46x_1 - 0.32x_2 + 1.18x_3 + 3.04x_4 - 0.45x_1x_2 - 0.81x_1x_3 - 0.43x_1x_4 - 0.31x_2x_3 \\ - 1.15x_2x_4 + 0.62x_3x_4 - 2.63x_1^2 - 3.59x_2^2 - 0.37x_3^2 - 1.65x_4^2 \tag{4}$$

where $x_1$ is the beating degree of cow dung, °SR; $x_2$ is the beating degree of flax straw, °SR; $x_3$ is quantitative, g/m²; and $x_4$ is the addition of flax straw fibre, %.

*3.4. Analysis of Variance of Regression Model*

The regression models (3) and (4) were analysed with ANOVA, and the results are shown in Table 3. From Table 3, it can be seen that the *p*-value of the quadratic regression model of the fibre paper film tensile strength $y_1$ is less than 0.0001, indicating that the model is highly significant. The coefficient of determination $R_1^2{}_{adj} = 0.9503$, which shows that the model is able to explain the change of 95.03% response value. The *p*-value of the missing item is 0.8137 (>0.05), and the missing item is not significant. The coefficient of determination $R_1^2 = 0.9751$, the coefficient of variation is 4.97% (<10%), and the signal-to-noise ratio is 21.451 (>4), which shows that the model fits reality well. The fitting reliability and precision are high, which shows that the quadratic regression equation fitted by the model is consistent with the fact and can correctly reflect the relationship between the paper tensile strength $y_1$ and the various factors $x_1$, $x_2$, $x_3$, and $x_4$. Regression models can effectively predict experimental results. Among them, the primary terms $x_2$, $x_3$ and $x_4$ are highly significant; the interaction term $x_2x_3$ is highly significant and $x_2x_4$ is significant. The quadratic terms $x_1^2$, $x_2^2$ and $x_4^2$ have highly significant effects, while the others are not significant.

At the same time, the *p*-value of the quadratic regression model of paper tear strength $y_2$ is less than 0.0001, which indicates that the model is highly significant. The corrected coefficient of determination $R_2^2{}_{adj} = 0.9692$, which shows that the model can explain the change of 96.92% response value. The *p*-value of the missing item is 0.5054 (>0.05), and the missing item is not significant. The coefficient of determination $R_2^2 = 0.9846$, the coefficient

of variation is 3.35% (<10%), and the signal-to-noise ratio is 30.838 (>4), which shows that the model is well fitted with the actual situation. The fitting reliability and precision are high, which shows that the quadratic regression equation fitted by the model is consistent with the real situation, and can correctly reflect the relationship between paper tear strength $y_2$ and the various factors $x_1$, $x_2$, $x_3$, and $x_4$. The regression models can be better predicted for the experimental results. Among them, the primary terms $x_1$, $x_3$, and $x_4$ have highly significant effects. The interaction items $x_1x_3$ and $x_2x_4$ have highly significant effects, and $x_3x_4$ has significant effects. The quadratic terms $x_1{}^2$, $x_2{}^2$, and $x_4{}^2$ have highly significant effects, while the others are not significant.

**Table 3.** Regression Model Analysis of Variance.

| Variance Source | Freedom | Taking Tensile Strength as the Response Value | | | | Taking Tear Strength as the Response Value | | | |
|---|---|---|---|---|---|---|---|---|---|
| | | Sum of Squares | Mean Square | F Ratio | *p* Ratio | Sum of Squares | Mean Square | F Ratio | *p* Ratio |
| Model | 14 | 57.69 | 4.12 | 39.12 | <0.0001 | 258.55 | 18.47 | 63.74 | <0.0001 |
| $x_1$ | 1 | 0.095 | 0.095 | 0.91 | 0.3569 | 2.59 | 2.59 | 8.92 | 0.0098 |
| $x_2$ | 1 | 6.59 | 6.59 | 62.66 | <0.0001 | 1.22 | 1.22 | 4.21 | 0.0593 |
| $x_3$ | 1 | 6.47 | 6.47 | 61.53 | <0.0001 | 16.82 | 16.82 | 58.04 | <0.0001 |
| $x_4$ | 1 | 17.26 | 17.26 | 164.16 | <0.0001 | 110.66 | 110.66 | 381.92 | <0.0001 |
| $x_1x_2$ | 1 | 0.0000 | 0.0000 | 0.0002 | 0.9879 | 0.78 | 0.78 | 2.70 | 0.1228 |
| $x_1x_3$ | 1 | 0.0081 | 0.0081 | 0.077 | 0.7854 | 2.63 | 2.63 | 9.07 | 0.0093 |
| $x_1x_4$ | 1 | 0.0081 | 0.0081 | 0.077 | 0.7854 | 0.73 | 0.73 | 2.52 | 0.1345 |
| $x_2x_3$ | 1 | 1.13 | 1.13 | 10.79 | 0.0054 | 0.39 | 0.39 | 1.35 | 0.2639 |
| $x_2x_4$ | 1 | 0.51 | 0.51 | 4.86 | 0.0447 | 5.27 | 5.27 | 18.19 | 0.0008 |
| $x_3x_4$ | 1 | 0.15 | 0.15 | 1.45 | 0.2490 | 1.54 | 1.54 | 5.32 | 0.0369 |
| $x_1{}^2$ | 1 | 22.84 | 22.84 | 217.29 | <0.0001 | 44.96 | 44.96 | 155.17 | <0.0001 |
| $x_2{}^2$ | 1 | 1.46 | 1.46 | 13.86 | 0.0023 | 83.78 | 83.78 | 289.14 | <0.0001 |
| $x_3{}^2$ | 1 | 0.13 | 0.13 | 1.27 | 0.2786 | 0.90 | 0.90 | 3.11 | 0.0998 |
| $x_4{}^2$ | 1 | 2.47 | 2.47 | 23.45 | 0.0003 | 17.55 | 17.55 | 60.57 | <0.0001 |
| Residual | 14 | 1.47 | 0.11 | | | 4.06 | 0.29 | | |
| Misfitting term | 10 | 0.84 | 0.084 | 0.53 | 0.8137 | 2.97 | 0.30 | 1.10 | 0.5067 |
| Error | 4 | 0.64 | 0.16 | | | 1.08 | 0.27 | | |
| Total deviation | 28 | 59.17 | | | | 262.60 | | | |
| The adjustment of the model determines the coefficient | | $R_1{}^2 = 0.9751$; $R_1{}^2{}_{adj} = 0.9503$ | | | | $R_2{}^2 = 0.9846$; $R_2{}^2{}_{adj} = 0.9692$ | | | |

Note: $p < 0.05$, with significant effect; $p < 0.01$, the effect is highly significant.

ANOVA was used to estimate the importance of each factor on the output response value and evaluate the applicability of the proposed model. Figure 5 is the contour map of the experimental data and predicted values for the beating degree of cow manure, the beating degree of flax straw, the weight of paper, and the amount of flax straw fibre added. The results indicate that the predicted values of the model are in good agreement with the experimental values [32].

*3.5. The Analyses of the Primary and Secondary Effect of Various Factors on the Tensile and Tear Strength of the Fibre Paper Films*

Regarding the calculation method of the importance of each factor in multiple quadratic regression, for the quadratic regression equation established from the experimental data, the test results of the coefficients of the quadratic equation can be used to determine the degree to which the factors act on y.

The quadratic regression model determines the importance of each effecting factor on the response value:

$$y = b_0 + \sum_{j=1}^{m} b_j x_j + \sum_{i \le j=1}^{m} b_{ij} x_i x_j + \sum_{i=1}^{m} b_i x_i^2 \tag{5}$$

where $b_0$, $b_j$, $b_{ij}$, and $b_i$ represent the constant term, linear term coefficient, interactive term coefficient, and quadratic term coefficient, respectively, and $x_i$ and $x_j$ represent the i and j factors, respectively.

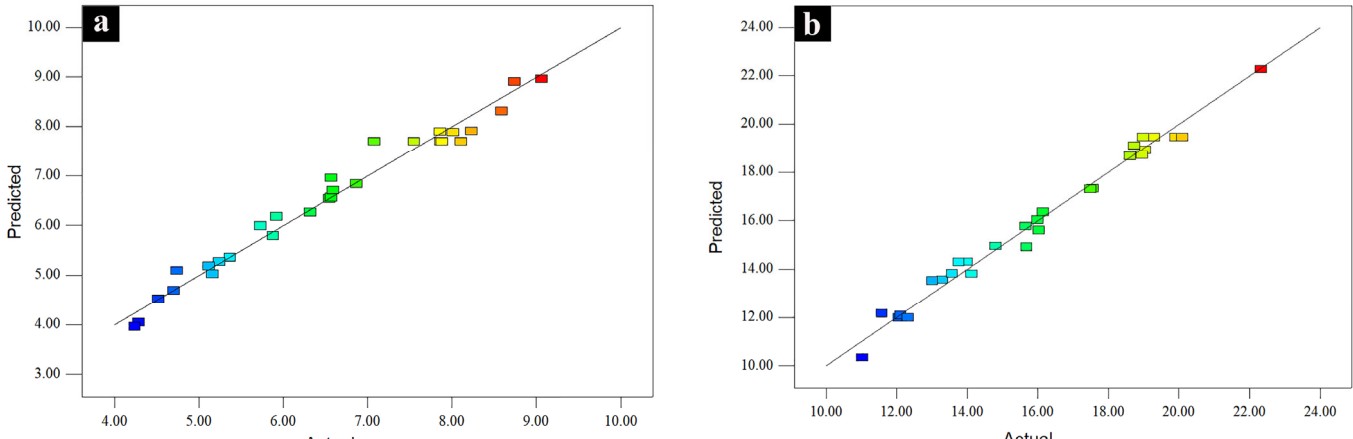

**Figure 5.** Equality graphs of predicted values against experimental data: (**a**) Tensile strength. (**b**) Tear strength.

The factors that affect the tensile strength of paper are $x_4 > x_3 > x_2 > x_1$, and the factors that affect the tear strength are $x_4 > x_3 > x_2 > x_1$. In summary, the order of effecting factors on the strength of the fibre paper film is from the largest to the smallest, which is the amount of flax straw fibre added, the paper basis weight, the beating degree of flax straw fibre, and the beating degree of cow dung fibre.

*3.6. The Effect Law of Various Factors on the Tensile and Tear Strength of the Fibre Paper Film*

3.6.1. The Effect Law on Tensile Strength

From Figure 6a, it can be seen that when the beating degree of flax straw fibre is fixed at a certain level, and the beating degree of cow dung fibre ranges from 33 °SR to 49 °SR, the tensile strength of the fibre paper film shows a trend of first increasing and then decreasing. When the beating degree of cow dung fibre is at the 0 level, the tensile strength reaches a peak; below the 0 level, the tensile strength increases with the beating degree of cow dung fibre; above the 0 level, the tensile strength decreases with the beating degree of cow dung fibre. When the beating degree of cow dung fibre is at a certain level, and the beating degree of flax straw fibre ranges from 81 °SR to 89 °SR, the tensile strength of paper tends to increase, and the changing trend below the 0 level is slightly greater than that of the 0 level. This is because at low beating degrees, the length of two types of fibres is longer, the brooming degree of flax straw fibre is lower, and the bonding among fibres is worse, which leads to loose paper film structure and uneven thickness, so the strength is lower. As the beating degree increases, the fibre length continues to decrease, and the tensile strength of the fibre paper film significantly improves. However, when the beating degree of two types of fibres exceeds a certain value, the structure of a single fibre is destroyed, and the bonding ability among fibres decreases, resulting in a slower trend in the tensile strength of the fibre paper film.

From Figure 6b, it can be seen that as the beating degree of flax straw fibre and the paper basis weight increase, the trend of the tensile strength of the fibre paper film changes slowly. When the paper basis weight is fixed at a certain level and the beating degree of flax straw ranges from 81 °SR to 89 °SR, the tensile strength of the fibre paper film shows a trend of first increasing and then decreasing. When the beating degree of flax straw fibre is at the 0 level, the tensile strength reaches a peak. Below the 0 level, the tensile strength increases with the increase in the beating degree of flax straw fibre. Above the 0 level, the tensile strength shows a decreasing trend with the increase in the beating degree

of flax straw fibre. This is because as the beating time increases, the temperature of the pulp continuously increases, and the flax straw fibres undergo swelling and brooming. The bonding effect among fibres is enhanced, and the comprehensive force is enhanced, resulting in an increase in the tensile strength of the fibre paper film. However, when the beating degree exceeds a certain level, the single-fibre structure is destroyed, leading to a decrease in the strength of the fibre paper film andresulting in a downward trend in tensile strength. When the beating degree of flax straw fibre is fixed at a certain level, the trend of increasing the tensile strength of the fibre paper film is not significant with the increase in the paper basis weight. This is because as the paper basis weight increases, the thickness of the fibre paper film also increases. According to Equation (1), the trend of changing the tensile strength of the fibre paper film is relatively insignificant for the increase in the paper basis weight.

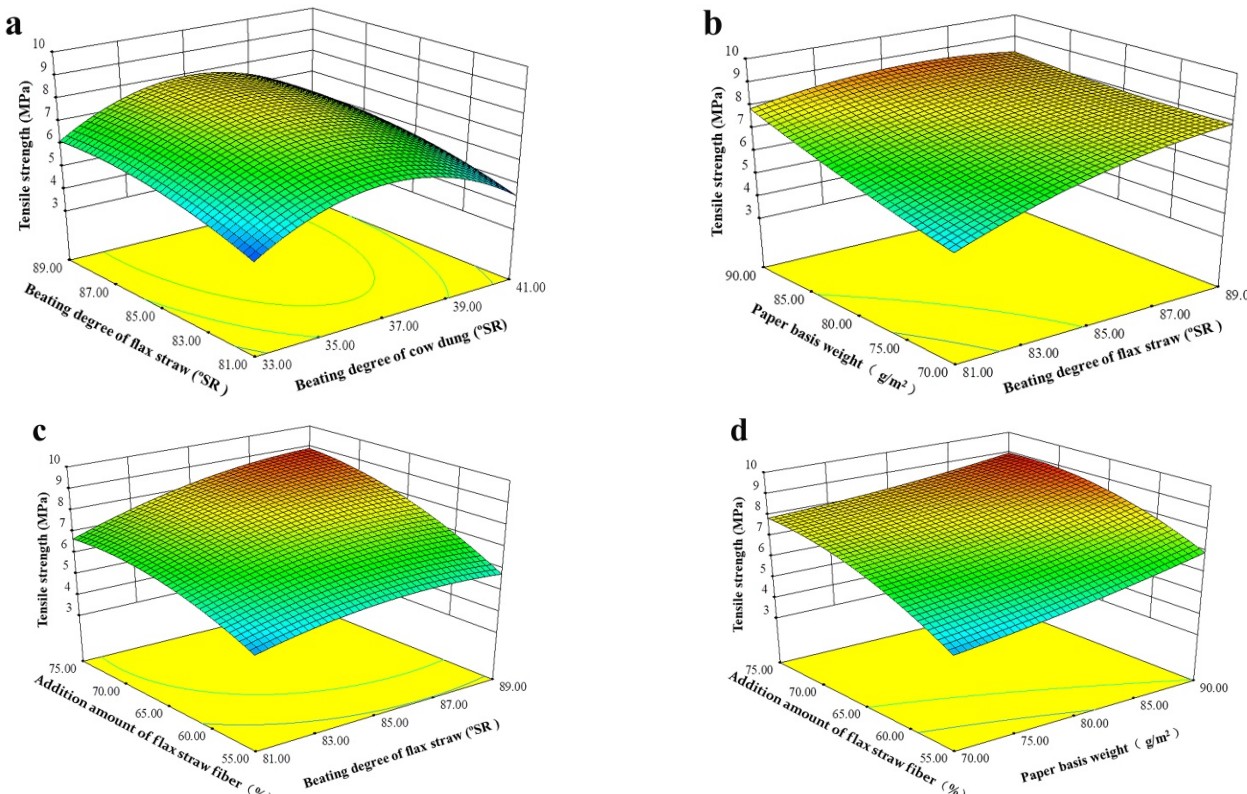

**Figure 6.** Response surface for effects of various factors on the tensile strength. (**a**) The effect of the beating degree of the cow dung and flax straw fibres on the tensile strength of fibre paper film. (**b**) The effect of the beating degree of the flax straw fibre and the paper basis weight on the tensile strength of fibre paper film. (**c**) The effect of the beating degree and the addition of flax straw fibre on the tensile strength of fibre paper film. (**d**) The effect of the addition of flax straw fibre and the paper basis weigh on the tensile strength of fibre paper film.

As can be seen from Figure 6c, when the beating degree of flax straw fibre is fixed at a certain level, and the addition of flax straw fibre ranges from 55% to 75%, the tensile strength of the fibre paper film shows a gradual increasing trend. When the addition of flax straw fibre is below the 0 level, the tensile strength shows an obvious change, and when it is above the 0 level, the changing trend is slow. When the addition of flax straw fibre is fixed at a certain level, as the beating degree of flax straw fibre increases, the tensile strength of the fibre paper film also increases. The trend of tensile strength change is more obvious below the 0 level, and the trend of tensile strength change is weakened above the 0 level. This is because as the beating degree of flax straw fibre increases, the brooming degree of the flax straw fibre increases, and the fibres are evenly dispersed. However, the structure of

a single fibre is destroyed, and the bonding effect among fibres decreases, resulting in a weakened trend in tensile strength. With the increase in flax straw fibre content, the fibre content per unit area of the fibre paper film will increase, so the tensile strength of the fibre paper film as a whole shows an increasing trend under the comprehensive force among fibres. From Figure 3, it can be observed that the tensile strength is the highest when the amount of flax straw fibre added is 75% and the beating degree of flax straw fibre is 89 °SR. From the response surface, it can be seen that the results are consistent with the analysis of variance.

It can be seen from Figure 6d that the tensile strength of paper increases with the increase in flax straw fibre content and the paper basis weight from a low level to a high level. Moreover, when the paper basis weight is above the 0 level, the changing trend of tensile strength slows down when the flax straw fibre content is high, because with the increase in the paper basis weight and flax straw fibre content, the flax straw fibre content per unit area of paper increases, reflecting a higher tensile stress, but at the same time, the thickness of the fibre paper film increases. According to Equation (1), the trend of increasing the tensile strength of the fibre paper film is relatively slow.

### 3.6.2. The Effect Law on Tear Strength

The effect of the beating degree of flax straw fibre and cow dung fibre on the tear strength of the fibre paper film is shown in Figure 7a when other factors are at the 0 level. When the beating degree of flax straw fibre is fixed at a certain level, the tear strength of the fibre paper film shows a trend of first increasing and then decreasing with the increase in cow dung fibre beating degree. When the beating degree of cow dung is below the 0 level, the tear strength shows an increasing trend with the increase in cow dung beating degree. And above the 0 level, the tear strength of the fibre paper film decreases with the increase in the beating degree of cow dung. When the beating degree of cow dung is fixed at a certain level, the tear strength of the fibre paper film first increases and then decreases with the increase in the beating degree of flax straw fibre. This is because when the beating degree of the two types of pulps is low, the fibre dimension is relatively coarse, and the flax straw fibres are intertwined, resulting in uneven fibre distribution and a poor bonding effect among fibres. As a result, the local tearing strength of the fibre paper film is weak, which affects the strength of the entire paper film. As the beating degree of the two types of pulps increases, the tear strength increases, and the bonding effect among fibres becomes more significant. As the beating time increases, when the beating degree exceeds a certain value, the structure of a single fibre is destroyed, and the fibre length becomes shorter, resulting in a decrease in the bonding force among fibres and a decrease in the overall fibre strength. When the fibre paper film is torn, more fibres are pulled off rather than broken, resulting in a decrease in the tear strength of the fibre paper film.

From Figure 7b, it can be seen that the tear strength of the fibre paper film increases with the increase in the paper basis weight. When the beating degree of cow dung fibre is above the 0 level, the changing trend of tear strength is smaller than when the beating degree of cow dung fibre is below the 0 level. This is because when the beating degree of cow dung fibre is low, the fibres are relatively coarse, unevenly dispersed, and the bonding effect among fibres is not tight, so the tear strength is lower. As the beating degree increases, the binding among fibres becomes tighter, and the tear strength increases accordingly. However, when the beating degree of torn cow dung exceeds a certain value, the structure of a single fibre is destroyed, causing a decrease in fibre strength. As the paper basis weight increases, the number of fibres per unit area increases. Under the combined force of fibres, the tear strength of the fibre paper film shows an increasing trend.

From Figure 7c, it can be seen that the tear strength of the fibre paper film gradually increases with the increase in the paper basis weight and the addition of flax straw fibres. With the increase in the paper basis weight, the tear strength shows an obvious increase when the amount of flax straw fibre is large. When the paper basis weight and flax straw fibre are above the 0 level, the overall increase trend in tear strength slows down. This

is because with the increase in the paper basis weight and flax straw fibre quantity, the number of flax straw fibres per unit area increases, and the tear force increases. At the same time, the fibre paper film thickness increases. From Figure 7c of the response surface, it can be seen that the maximum effect of the paper basis weight and flax straw fibre addition on the fibre paper film tear strength appears in the basis weight of 90 g/m$^2$ and flax straw fibre addition of 75%.

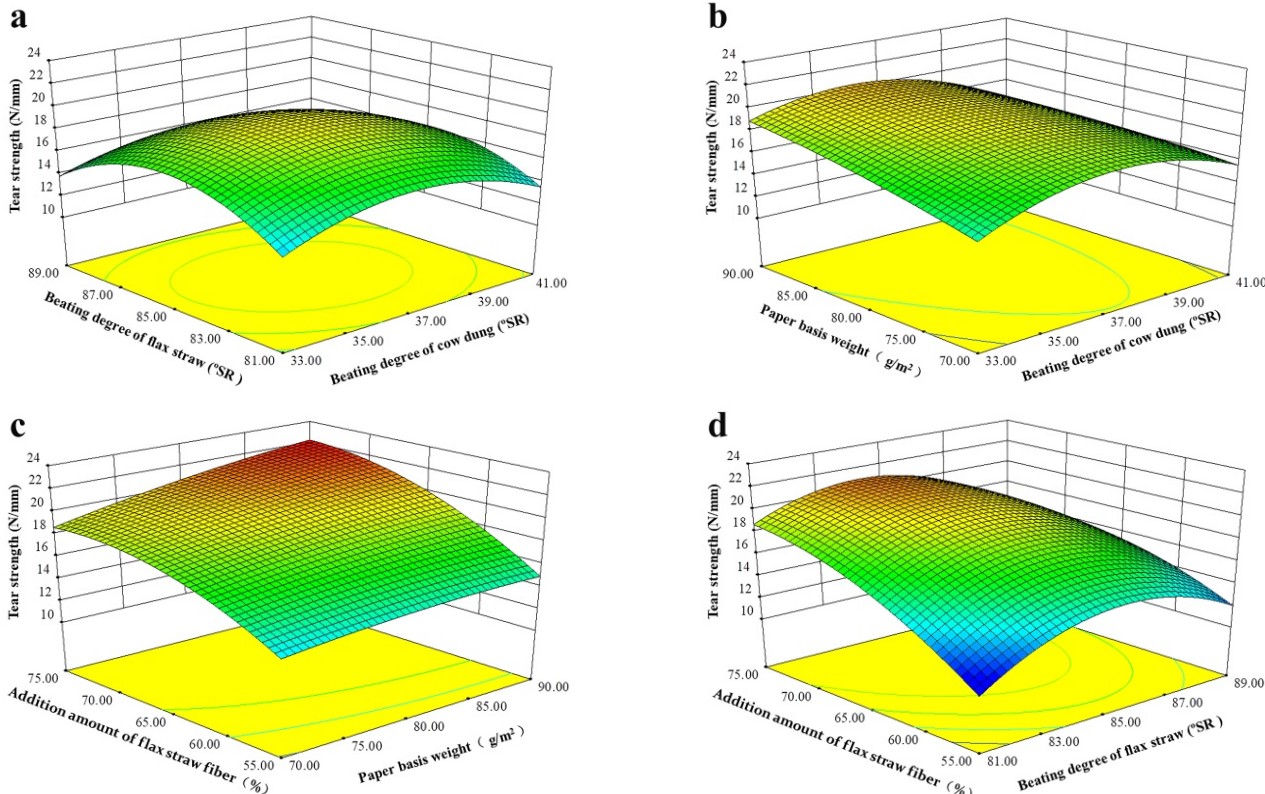

**Figure 7.** Response surface for effects of factors on tear strength. (**a**) The effect of the beating degree of the cow dung and flax straw fibres on the tear strength of fibre paper film. (**b**) The effect of the beating degree of the cow dung and the paper basis weight on the tear strength of fibre paper film. (**c**) The effect of the addition of flax straw fibre and the paper basis weigh on the tear strength of fibre paper film. (**d**) The effect of the beating degree and the addition of flax straw fibre on the tear strength of fibre paper film.

From Figure 7d, it can be found that when the beating degree of flax straw fibre is below the 0 level, the fibre paper film tearing strength increases with the increase in flax straw fibre content, and when the beating degree of flax straw fibre is above the 0 level, the paper tearing strength first increases and then decreases with the increase in flax straw fibre content, and the decrease is slight. This is because when the beating degree is low, the degree of brooming of flax straw fibre is lower, and a large number of fibres exist in the form of flocs, which are unevenly dispersed, affecting the tear strength of the fibre paper film. With the increase in beating degree, the tear strength of the fibre paper film is significantly improved. However, when the beating degree exceeds a certain value, the strength of a single fibre is destroyed, and the fibres are fine, resulting in a decrease in the bonding force among fibres. After the fibre paper film is subjected to force, more fibres are pulled off, which leads to a decrease in tear strength. With the increase in flax straw fibre content, the fibre content in the unit paper area increases, so the tear strength is enhanced under the comprehensive force of fibres. Through the analysis of the response surface shown in Figure 7d, it can be seen that the effect of flax straw fibre content on paper tear

strength is more significant than that of flax straw fibre beating degree. The maximum value appears at 85 °SR and flax straw fibre content is 75%.

### 3.7. Determination of Optimal Parameters and Experimental Verification

According to the analysis of the above test results, to further improve the mechanical strength of the fibre paper film, under the horizontal constraints of various test factors, the maximum tensile strength and tear strength of the fibre paper film were taken as optimization indexes. The quadratic regression equation of all factors of performance indexes was established to optimize the target and determine the optimal parameters.

$$\begin{cases} \max y_1(x_1, x_2, x_3, x_4) \\ \max y_1(x_1, x_2, x_3, x_4) \\ -1 \leq x_1 \leq +1 \\ -1 \leq x_2 \leq +1 \\ -1 \leq x_3 \leq +1 \\ -1 \leq x_4 \leq +1 \end{cases} \tag{6}$$

The regression equation models (3) and (4) are optimally solved by the optimizer in the programme Design-Expert 8.0.6. The optimized test indexes are a tensile strength of 8.93 MPa and tear strength of 22.51 N/mm, and the best parameter combination is the beating degree of cow dung of 36.47 °SR, the beating degree of flax straw fibre of 84.86 °SR, the basis weight of 90.00 $g/m^2$, and the addition amount of flax straw fibre of 74.78%.

According to the optimal process parameters and the principle of low production cost, to make the fibre paper film sample, the beating degree of cow dung fibre is 37 °SR, the beating degree of flax straw fibre is 85 °SR, the paper basis weight is 80 $g/m^2$, and the addition amount of flax straw fibre is 65%. After being stabilized for 24 h in the laboratory, the performance evaluation indexeswere determined, and eight parallel tests were conducted. The tensile strength was about 8.26 MPa, and the tear strength was about 19.91 N/mm. The verification test results show that the optimization results are correct and reliable.

In this study, two types of agricultural and animal husbandry waste materials, cow dung and flax straw, were used as raw materials for the preparation of fibre paper films. Based on the optimal process parameters and the principle of low production cost, to better utilize cow dung and flax straw, the beating degree of cow dung fibre was determined to be 37 °SR, the beating degree of flax straw fibre was 85 °SR, the paper basis weight was 80 $g/m^2$, and the addition amount of flax straw fibre was 65%. The fibre paper film samples were manufactured. After standing in a laboratory environment for 24 h, the samples were subjected to eight parallel tests to determine various performance evaluation indicators. The tensile strength was about 8.26 MPa, and the tear strength was about 19.91 N/mm, which was close to the fibre paper film strength under the optimal parameters. The strength was comparable to the film strength studied by Jeetah et al. [33] and Koray Gulso et al. [30], and the validation test results showed that the optimization results were correct and reliable.

In the existing related reports, there are fewer reports on the preparation of fibre paper films using a mixture of cow dung and other straws. In this study, in order to determine the feasibility of these two kinds of materials (cow dung and flax straw) as raw materials to prepare fibre paper films, we mainly studied and analysed the preliminary and the most simplified preparation process of the fibre paper film, optimized the basic parameters of the preparation process, and determined a set of reliable preparation process parameters. The tensile strength of 8.26 MPa and the tear strength of 19.91 N/mm were the properties of the fibre paper film without any functional additives or modifying reagents added during the preparation process. So compared with other paper films reported which were modified or included additives, there is still a certain gap in performance. Based on this work, further research can focus on the improvement of the properties of paper film by selecting various pre-treatment methods, additives, or modifications.

## 4. Conclusions

Mulch is widely used in agriculture as a means of production. Using agricultural and livestock waste as raw materials to prepare pure green fibre paper film shows potential in sustainable agricultural development. This exploration of the feasibility of comprehensively utilizing flax straw and cow dung to produce fibre paper film is a meaningful study, which can provide a reference for further research on fibre paper film by mixing flax straw and cow dung as raw materials. This article analyses the preparation process of the fibre paper film and screens the main effecting factors of the fibre paper film strength based on single-factor experiments. According to the Box–Behnken experimental design principle, the process parameters of the fibre paper film preparation were optimized using four factors and three levels response surface analysis method. The results showed that the primary and secondary order of factors affecting the mechanical strength of the fibre paper film was the amount of flax straw fibre added, the beating degree of flax straw fibre, the basis weight of paper, and the beating degree of cow dung fibre. By establishing a quadratic regression model, with the aim of maximizing the tensile strength and tear strength of the fibre paper film, the optimal process parameters were as folloes: the beating degree of the cow dung fibre was 37 °SR, the beating degree of the flax straw fibre was 85 °SR, the paper basis weight was 80 $g/m^2$, and the addition amount of flax straw fibre was 65%. The validation test shows that the tensile strength was 8.26 MPa, and the tear strength was 19.91 N/mm. This experimental data can serve as a reference for subsequent research in its degradability and play a certain role in the study of plastic film replacement products.

**Author Contributions:** Author Contributions: Conceptualization, W.Z., F.Z. (Fengwei Zhang) and L.L.; Investigation, X.C.; methodology, F.D. and L.S.; Data curation, X.C., F.Z. (Fangyuan Zhang) and X.S.; writing—original draft preparation, X.C.; writing—review and editing, L.L. and F.D.; Project administration, F.Z. (Fengwei Zhang), L.L. and L.S.; Funding acquisition, L.L. and F.Z. (Fengwei Zhang). All authors have read and agreed to the published version of the manuscript.

**Funding:** This research was funded by Gansu Agricultural University Talent Introduction Program (GAU-KYQD-2019-16), Special Project for Science and Technology Commissioner of Gansu Provincial Department of Science and Technology (23CXGA0072), Gansu province college students' innovative training project (S202310733027), National Natural Science Foundation of China (No. 32260432) and Gansu Provincial Science and Technology Plan (Key R&D Plan) Project (21YF5NA092).

**Institutional Review Board Statement:** Not applicable.

**Informed Consent Statement:** Not applicable.

**Data Availability Statement:** Data are contained within the article.

**Conflicts of Interest:** The authors declare no conflict of interest.

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
