# Peer review of "Optimization of the Green Fibre Paper Film Preparation Process Based on Box–Behnken Response Surface Methodology"

_coatings, doi:10.3390/coatings13122025_

Round 1

Reviewer 1 Report

Comments and Suggestions for Authors

This manuscript deals with an approach for the optimization of fiber paper film preparation by mixing cow dung and flax straw film without using any additive. This paper is recommended to be accepted for publication after some revision on the basis of comments below.

COMMENTS

1.

Tensile strength and tear strength of 8.257 MPa and 19.914 N/mm in the Abstract and corresponding data in Table 3 have too many decimal digits. The error of the applied measurements is much higher. One, maximum two digits can be accepted on the one hand. The error of these measurements, have to be provided.

2.

In line 30, China is mentioned as “a big agricultural country” for using plastic films for water-saving purposes. Every country with agriculture uses such methods, so this sentence is suggested to be replaced as follows:  “Plastic films are widely used as water-saving measures in agriculture worldwide [3].”

3.

In the Experimental, lines 128-130, the authors write that the “cow dung raw materials was soaked and cleaned”, but no any details are provided on the materials and amounts which were used to soak and clean the cow dung. These must be described in details in order to provide exact data for the readers if they want to repeat these experiments.

4.

In line 132-133, the authors write that the flax straw was cooked at 160 C, but no details on the experimental equipment and added substances, e.g. solvent (water ?) are provided. The authors have to describe the cooking process in details.

5.

In line 151, soaking of the dry pulp is mentioned. However, again, no any detail is provided about the soaking material (solvent) and the quantities of the dry pulp and soaking solvent. These data should be added.

6.

In Figure 1b, the meaning of R20, R25, R26.5 and R12.5 is absolutely unclear. This must be provided either in the text or in the figure caption.

7.

What the authors mean on “The addition amount of flax draw fibre” in Table 2? This should be described and explained in the text.

8.

The error bars on the top of the column diagrams in Figure 2 must be provided.

9.

Comparing the data in Figures 2a, 2b, 2c and 2d, it is evident that the the amount of the added flax straw fiber and the paper basis weight have in this order the higher effect on the mechanical properties. Therefore, multivariable regression of the data seems useless. The authors should describe in details the advantages of carrying out such regression, when the experimental data show obvious correlation.

10.

In section 3.6, the authors must describe in a clear way how the surfaces were constructed, calculated in Figures 4 and 5

Reviewer 2 Report

Comments and Suggestions for Authors

The abstract can be made more attractive by using more quantitative data. It is suggested to provide more quantitative results.

The use of abbreviations is allowed only if they are fully defined in the first reference. These items must be observed in the abstract.

The research method section is very long and should be presented more briefly.

Use the following resources to deepen the introduction and discussion. Shape memory performance assessment of FDM 3D printed PLA-TPU composites by Box-Behnken response surface methodology. Toughening PVC with Biocompatible PCL Softeners for Supreme Mechanical Properties, Morphology, Shape Memory Effects, and FFF Printability.

The way of referencing in the introduction should be modified. The use of general sentences with more than four references can be seen in the introduction (Lines 33, 38, and etc ).

How has the reproducibility of these results been checked? Specify the number of tensile tests conducted. Error bar added to mechanical properties results.

The results section is well organized and categorized. However, some parts report the results, which require corrections and deepening the analysis and discussion.

The conclusion needs rewriting. In the conclusion section, a summary of the purpose of the research, innovation, and research method should be presented before presenting the highlights.

Microstructure images (SEM) do not have labels and scale bars. The use of labels can explain the explanation of this section more clearly.

Comments on the Quality of English Language

No comment.

Reviewer 3 Report

Comments and Suggestions for Authors

The manuscript reports the experimental testing results of the physical and mechanical properties of green paper made from cow dung waste and flax fibres. The manuscript is well-written and provides helpful results. This paper suits the Special issue "Bio-Based and Bio-Inspired Polymers and Composites" in Coatings. The following comments should be addressed to improve the manuscript before it is accepted for publication:

Introduction

Although the introduction is well-written, the authors should emphasize in the last paragraph of the introduction what is new and unique in this work that has not been done before, i.e., what is the novelty? The authors should emphasize this with respect to their previous work so it is clear to the reader what the novelty is concerning the manufacturing and testing of paper from cow dung.

Section 2

-Please add in Section 2.2 a schematic/diagram that visually explains the manufacturing process of the cow dung/flax fibre paper specimens.

Section 3

-At the end of section 3, please add a Table and compare some of the physical and mechanical properties of the paper from this work to the properties of other paper made from cow dung or other agricultural waste, similar to Table 4 in the author's previous publication; however please add a more extended discussion of the paper presented in this work and its advantages and disadvantages.

Round 2

Reviewer 3 Report

Comments and Suggestions for Authors

The authors have satisfactorily addressed Comments 1 and 2; however, Comment 3 still needs to be addressed correctly. The response to comment 3 given in the author's reply (see below) is appropriate. The most important points of response 3 (see below) should be added at the end of Section 3 as a discussion that shows the limitations of the work (considering this is preliminary work as stated by the authors) and the plans for further research to improve the paper film presented in the manuscript.

Comments 3: At the end of section 3, please add a Table and compare some of the physical and mechanical properties of the paper from this work to the properties of other paper made from cow dung or other agricultural waste, similar to Table 4 in the author's previous publication; however please add a more extended discussion of the paper presented in this work and its advantages and disadvantages. 

Response 3: Thank you for pointing this out. We agree with your comments. But, among the existing related reports, there are fewer reports on the preparation of fibre paper film using a mixture of cow dung with other straws. In this study, in order to determine the feasibility of these two kinds of materials (cow dung and flax straws) as raw materials to prepare fibre paper film, we mainly studied and analyzed the preliminary and the most simplified preparation process of paper film, optimized the basic parameters of the preparation process, determined a set of reliable preparation process parameters. The tensile strength of 8.26 MPa and the tear strength of 19.91 N/mm were the properties of the fibre paper film without any functional additives or modifying reagents added during the preparation process. Based on this work, further research can focus on the improvement of the properties of paper film by selecting various pre-treatment methods, additives, or modifications. So compared with other paper films reported which were modified or with additives, there is still a certain gap in performance. Therefore, we feel that a comparison between a preliminary prepared product and a paper film that has been treated and modified is an undesirable choice.

Author Response

We would like to thank you for your consideration of our manuscript and thank reviewers for their detailed and constructive comments. Please find our point-by-point responses in the following pages and changes in the manuscript highlighted in red. We hope you and the reviewers will find the responses and changes satisfactory.

With best regards,

Xiaoqing Cao, Lu Li, Fei Dai

Optimization of the Green Fibre Paper Film Preparation Process Based on Box-Behnken Response Surface Methodology

coatings-2720492

Comments: The authors have satisfactorily addressed Comments 1 and 2; however, Comment 3 still needs to be addressed correctly. The response to comment 3 given in the author's reply (see below) is appropriate. The most important points of response 3 (see below) should be added at the end of Section 3 as a discussion that shows the limitations of the work (considering this is preliminary work as stated by the authors) and the plans for further research to improve the paper film presented in the manuscript.

Response: Thank you for pointing this out. We agree with your comments. Therefore, we have added the discussion at the end of section 3 and marked it in red.

Change: In the existing related reports, there are fewer reports on the preparation of fibre paper film using a mixture of cow dung and other straws. In this study, in order to determine the feasibility of these two kinds of materials (cow dung and flax straws) as raw materials to prepare the fibre paper film, we mainly studied and analyzed the preliminary and the most simplified preparation process of the fibre paper film, optimized the basic parameters of the preparation process, determined a set of reliable preparation process parameters. The tensile strength of 8.26 MPa and the tear strength of 19.91 N/mm were the properties of the fibre paper film without any functional additives or modifying reagents added during the preparation process. So compared with other paper films reported which were modified or with additives, there is still a certain gap in performance. Based on this work, further research can focus on the improvement of the properties of paper film by selecting various pre-treatment methods, additives, or modifications.
